# Clinical Characteristics and Survival Analysis in a Small Sample of Older COVID-19 Patients with Defined 60-Day Outcome

**DOI:** 10.3390/ijerph17228362

**Published:** 2020-11-12

**Authors:** Agnieszka Neumann-Podczaska, Michal Chojnicki, Lukasz M. Karbowski, Salwan R. Al-Saad, Abbas A. Hashmi, Jerzy Chudek, Slawomir Tobis, Sylwia Kropinska, Iwona Mozer-Lisewska, Aleksandra Suwalska, Andrzej Tykarski, Katarzyna Wieczorowska-Tobis

**Affiliations:** 1Geriatric Unit, Department of Palliative Medicine, Poznan University of Medical Sciences, 61-245 Poznan, Poland; lx.karbowski@gmail.com (L.M.K.); Salwan1996@gmail.com (S.R.A.-S.); abbashashmi@icloud.com (A.A.H.); skropins@ump.edu.pl (S.K.); kwt@tobis.pl (K.W.-T.); 2Department of Biology and Environmental Protection, Poznan University of Medical Sciences, 60-806 Poznan, Poland; mchojnicki@gmail.com (M.C.); iwonalisewska@poczta.onet.pl (I.M.-L.); 3Department of Infectious Diseases, Jozef Strus Hospital, 61-285 Poznan, Poland; 4Department of Internal Medicine and Oncological Chemotherapy, Medical University of Silesia, 40-027 Katowice, Poland; chj@poczta.fm; 5Occupational Therapy Unit, Department of Geriatric Medicine and Gerontology, Poznan University of Medical Sciences, 60-781 Poznan, Poland; slawomir@tobis.pl; 6Department of Mental Health, Chair of Psychiatry Poznan University of Medical Sciences, 60-572 Poznan, Poland; asuwalska@gmail.com; 7Department of Hypertensiology, Angiology and Internal Medicine, Poznan University of Medical Sciences, 61-848 Poznan, Poland; tykarski@o2.pl; 8Geriatric Outpatient Clinic, University Hospital of Lord’s Transfiguration, 61-245 Poznan, Poland

**Keywords:** COVID-19, SARS-CoV-2, elderly, independence, 60-day survival, prognosis

## Abstract

The older population is one of the most vulnerable to experience adverse outcomes of COVID-19. Exploring different clinical features that may act as detrimental to this population’s survival is pivotal for recognizing the highest risk individuals for poor outcome. We thus aimed to characterize the clinical differences between 60-day survivors and non-survivors, as well as analyze variables influencing survival in the first older adults hospitalized in Poznan, Poland, with COVID-19. Symptoms, comorbidities, complications, laboratory results, and functional capacity regarding the first 50 older patients (≥60 years) hospitalized due to COVID-19 were retrospectively studied. Functional status before admission (dependent/independent) was determined based on medical history. The 60-day survivors (*n* = 30/50) and non-survivors (*n* = 20/50) were compared across clinical parameters. The patients had a mean age of 74.8 ± 9.4 years. Overall, 20/50 patients died during hospitalization, with no further fatal outcomes reported during the 60-day period. The non-survivors were on average older (78.3 ± 9.7 years), more commonly experienced concurrent heart disease (75%), and displayed functional dependence (65%) (*p* < 0.05). When assessing the variables influencing survival (age, heart disease, and functional dependence), using a multivariate proportional hazards regression, functional dependence (requiring assistance in core activities of daily living) was the main factor affecting 60-day survival (HR, 3.34; 95% CI: 1.29–8.63; *p* = 0.01). In our study, functional dependence was the most important prognostic factor associated with mortality. Elderly with COVID-19 who required assistance in core activities of daily living prior to hospitalization had a three times increased risk to experience mortality, as compared to those with complete independence. Exploring geriatric approaches, such as assessment of functional capacity, may assist in constructing comprehensive survival prognosis in the elderly COVID-19 population.

## 1. Introduction

The novel SARS-CoV-2 virus has been shown to exceedingly differentiate between an array of populations based on several contrasting factors, with age being a common variable. As COVID-19 data from various countries continue to emerge, a substantial disparity in the disease severity and fatality rates between different age groups become increasingly evident. The World Health Organization (WHO) attributed 90% of all COVID-19 deaths to older adults, aged ≥60 years [1]. Other publications resonate with these statistics [2,3]. Richardson and colleagues [4] reported 5700 cases of COVID-19 in New York City and noted three in four deaths to be of patients ≥65 years of age. Epidemiological data for a host of European countries state a 15–100-fold higher risk of death in SARS-CoV-2 patients aged ≥65 years compared to those under the age of 65, in economically advanced countries [5]. Such alarming numbers warrant the establishment of older age as a key factor in the risk assessment and prognosis of COVID-19 and further necessitate particular attention to this age group.

In addition to high mortality rates in older patients, current studies have shown an increased severity of disease and likelihood for admission into intensive care for older patients who are hospitalized due to the SARS-CoV-2 virus [6]. Several factors may be associated with this increased severity of disease, including general and immune deterioration in physiological efficiency [7]. It is currently understood that a harsher course of COVID-19, in addition to the increased fatality rates, is more likely in those with pre-existing comorbidities, such as diabetes, cardiovascular disease (including hypertension), and respiratory tract diseases [6,8,9]. Few other prognostic factors have also been suggested, such as malignancy and chronic kidney disease [10,11,12]. Correspondingly, understanding clinical characteristics (such as symptoms, comorbidities, and functional capacity) affecting novel coronavirus survival prognosis and their role in the severity of COVID-19 course of disease is crucial in protecting older adults.

Identifying factors prompting a worsened course of COVID-19 and their correlation to survival rate can assist in early prediction of high-risk older patients. Gemes and colleagues demonstrated that an analysis of prognostic factors and comorbidities is capable of generating an estimation of a given country’s population at risk for a severe course of COVID-19 [13], highlighting the importance of investigating the cause of the unbalanced predilection of SARS-CoV-2 virus toward older patients. It is possible that methodical prognosticators capable of characterizing high risk older patients are still undeclared. It is therefore vital that risk factors and variables affecting the course of the disease continue to be identified and revised in order to develop strategies to safeguard and minimize fatalities from SARS-CoV-2 infection in older populations. A more precise prediction of outcomes and identification of those at risk of infection may also contribute to lowering the necessity of imposing harsh restriction for whole populations, thus easing the preventive measures for the less affected social groups.

We thus aimed to characterize the difference between 60-day survivors and non-survivors among hospitalized older adults and further investigate how those with increased risk of poor outcome can be identified as early as possible. Clinical data, including patient symptoms, comorbidities, complications, and functional capacity, were assessed to identify factors affecting outcome of SARS-CoV-2 infection in elderly patients. To the best of our knowledge, this paper presents the Central-Eastern European perspective of COVID-19 for the first time.

## 2. Method

Data from the first 50 older patients (≥60 years of age) hospitalized for COVID-19 in the Jozef Strus Hospital, Poznan, Poland, were extracted and retrospectively analyzed. The data include patient symptoms, complications, laboratory results, and functional capacity prior to hospitalization, with patient outcome, discharged or dead, known prior to analysis. All studied patients had a diagnosis of COVID-19 confirmed by a positive result of RT-PCR testing of a nasopharyngeal swab. The Vitassay^®^ assay was used at Cobas^®^ 2480 (Roche) system, and RNA isolation was performed using Geneproof. The inclusion criterion was hospitalization lasting at least 48 h.

In Poland, on 8 March, one hospital in each voivodeship (total of 16, preferably those with infectious wards) were transformed into monomial hospitals, dedicated entirely to the treatment of COVID-19, therein also the Jozef Strus Hospital in Poznan, Poland. Upon suspicion of COVID-19, community-dwelling patients were referred to laboratories (including the ones in dedicated COVID-19 hospitals) which performed RT-PCR tests for SARS-CoV-2. Gene ORF1 ab and *n* were tested with declared test sensitivity above 10 viral copies, as a measure to standardize testing. If tested positive and symptomatic, the patients were either hospitalized in monomial hospitals or—in milder cases—isolated at home with quarantine regulations and mandatory regular contact with medical staff over the phone. As for institutionalized adults (e.g., residents of nursing homes, care homes, or residential facilities) or patients hospitalized in non-monomial hospitals, the test was performed in the place of stay. If tested positive, these patients were transferred to a dedicated COVID-19 hospital; this was, notably, also the case for patients from palliative care units or hospices.

The first COVID-19 patient (aged ≥ 60) included in the study was hospitalized in the Jozef Strus Hospital on 12 March 2020 and the last on 5 April 2020. For all patients, demographic information (age, sex, and place of living: home/institution), clinical, laboratory, treatment, and outcome data were extracted from the in-hospital medical records. Measurements were based on clinical judgement and recorded by the attending physicians. Exacerbations of pre-existing chronic conditions were judged taking into account previous medical history and comorbidities of individual patients as compared to their health status and progression at COVID-19 hospitalization.

As far as clinical data are concerned, symptoms at admission were analyzed (mainly, fever, cough, dyspnea, and muscular pain); in addition, any new symptoms and exacerbations of pre-existing chronic clinical conditions were noted. Fever was defined as the axillary temperature above 38.0 °C while subfebrile condition was understood as a temperature of 37.1–38.0 °C. Additionally, the presence of cognitive impairment (including delirium) was taken into account. Cognitive impairment was determined based on standard clinical assessment procedure including physical assessment by attending clinicians upon admission to the hospital, as well as past medical history assessment.

The medical history of cardiovascular diseases (hypertension, congestive heart failure, ischemic heart disease, arrhythmias, and valvular diseases), chronic obstructive pulmonary disease (COPD) or asthma, diabetes, kidney or liver insufficiency, stroke, and dementia and cancer (current or within the last five years) was also retrieved. Infectious disease specialists or internal medicine physicians and nurses’ reports included in hospital medical records were in combination the basis for assessments.

On admission, the patients were additionally assigned a COVID-19 clinical status by specialized physicians in internal medicine or infectious disease, based on the report by WHO-China Joint Mission [14]: (1) (Mild) lab confirmed but no pneumonia present; (2) (Moderate) lab confirmed with pneumonia present; (3) (Severe) respiratory rate ≥ 30 per minute, dyspnea present, oxygen saturation in blood ≤ 93%; and (4) (Critical) required mechanical ventilation due to respiratory failure or intensive care due to severe deterioration of condition from organ failure.

If available, laboratory data analysis included white blood cells (WBC), lymphocytes (L) and neutrophils (Ne) count as well as red blood cells (RBC), hemoglobin, platelet count, lactate dehydrogenase (LDH), and inflammatory markers (C-reactive protein (CRP), procalcitonin (PCT), and interleukin 6 (IL-6)). Additionally, the functions of liver and kidney were assessed. Kidney function was considered abnormal when eGFR (estimated glomerular filtration rate) was below 60 mL/min/1.73 m^2^ [15]. As for liver function, abnormal status was recognized when the level of at least one enzyme was above its reference value. All laboratory analyses were based on blood samples retrieved from patients and tested in the central labs of the monomial hospitals.

Thereafter, the data of chest X-ray or CT (computed tomography) scans were screened for unilateral or bilateral pneumonia, the presence of pleural effusion, or the typical changes for COVID-19 (usually referred to as ground-glass opacities or consolidations). X-rays and CT scans were performed and reviewed by hospital radiologists and given further review by attending clinicians.

Treatment data analysis included the prescription of aminoquinoline derivatives; the following groups of antibiotics: penicillin, cephalosporin, macrolides, tetracycline, carbapenem-type antibiotics, glycopeptide antibiotics, oxazolidinones; and antiviral therapy, interleukin-6 receptor inhibitor, low-molecular heparin derivatives, and oral anticoagulants. Oxygen therapy was also taken into consideration—both non-invasive (i.e., mask) and invasive (mechanical ventilation).

In all patients, the outcome data (either death or discharge) were recorded. To declare recovery (and consequent discharge from the hospital), patients had to display negative results of two separate RT-PCR tests. Furthermore, the data of in-hospital survival were analyzed. For those discharged, their 60-day survival was assessed using the hospital’s system for the monitoring of mortality which is directly linked and updated simultaneously with the national social security body. Based on the date of admission and the date of discharge (or death), the length of hospitalization was calculated. We also verified if patients were previously hospitalized elsewhere for other reasons, after COVID-19 diagnosis.

Based on the medical history taken on admission, the functional status was assessed as: independent (when the patient was independent in all instrumental and basic activities of daily living, with the exception of mild urinary incontinence) or dependent (in all other cases). Partial dependence was defined as dependence in at least one instrumental activity of daily living and independence in all basic activities (e.g., a patient who was walking with sticks and was not able to do shopping) and total dependence was defined as the subject being dependent in at least one basic activity of daily living (e.g., able to transfer from the bed only with help). The assignment of functional status of patients was performed by two experienced geriatric professionals (members of EAMA (European Academy for Medicine of Ageing)). Subsequently, an infectious disease specialist from the dedicated COVID-19 hospital double-checked the consistency of assessment and, in the case of doubt, directly contacted the patient or the family/caregiver to clarify the status. Clear consensus was reached concerning the functional status of all the patients.

Due to the retrospective nature of the presented study, written consent by participants was not necessary. The presented project was approved by the Bioethical Committee of the Poznan University of Medical Sciences (Resolution Number KB-380/20).

### Statistical Analysis

Statistical analysis was performed using the STATISTICA13.0^®^ package (Tibco Software Inc. Palo Alto, CA, USA). To evaluate the normality of the distribution of the variables, the Shapiro–Wilk test was applied. Continuous data are presented as mean ± SD. Due to lack of normality of some of the variables, median and range were also presented. Continuous variables were compared using the Mann–Whitney test. Categorical variables were expressed as numbers (percentage) and compared with the χ^2^ test with the Yates correction applied due to small sample size.

Overall 60-day survival, which includes both in-hospital and out-of-hospital mortality, was compared by the Kaplan–Meier method and log-rank test. Multivariate Cox proportional hazard analyses were performed for factors significantly influencing the survival in simple models [16]. Variables found significantly different between surviving and non-surviving patients were included in the model. Adjustment for confounder was restricted due to limited sample size.

Statistical significance was set at *p* < 0.05.

## 3. Results

The study group consisted of 50 patients (35 males and 15 females) mostly (*n* = 29 (58%)) transferred from other non-COVID-19 hospitals, including two patients transferred from palliative care units. The mean age of the total group was 74.8 ± 9.4 years (median: 72.5; range: 61–99). The calculated person-years accumulated by the patients was 3.02 years.

The patients’ characteristics are presented in Table 1. On admission, increased body temperature was present in 29 (58%) subjects (including 14 subfebrile) and dyspnea in 17 (34%). They were the most common symptoms related to COVID-19. Cognitive impairment was present in 11 patients (22%), including six with delirium. There were two subjects with gastrointestinal symptoms (*n* = 2 (4%)). Only four (8%) patients presented no concomitant comorbidities, while *n* = 30 patients (60%) experienced two or more comorbidities. The majority had cardiovascular diseases (*n* = 40 (80%)), including *n =* 30 (60%) with hypertension. The COVID-19 clinical status of the older patients was as follows: *n* = 11 (22%) mild, *n* = 24 (48%) moderate, *n* = 11 (22%) severe, and *n* = 4 (8%) critical. Based on functional capacity, *n* = 20 (40%) patients were characterized as dependent before hospitalization, including *n* = 6 (12%) totally dependent.

RBC count, hematocrit, and hemoglobin in the studied population were, on average, slightly below normal values. The mean levels of urea, LDH, and inflammatory markers (CRP, PCT, and IL-6) were also observed to be above reference values (Table 2). Kidney insufficiency was diagnosed in 19 patients (38%). Liver enzymes were increased in *n =* 9 subjects (out of 38 in whom they were measured (24%)).

Radiographic imaging was performed in *n* = 41 (82%) of the patients, revealing bilateral lung involvement in *n* = 26 of those patients (64%), and unilateral in *n* = 7 (17%). In *n* = 8 (19%) of the older adults, no abnormalities were found in radiographic imaging. Consolidations were the most commonly described findings (*n* = 38 (76%)), followed by occasional ground-glass opacities (GGO) (*n* = 10 (24%)) and pleural effusion (*n* = 9 (22%)).

A combination treatment regimen (more than one type of pharmacotherapy) was employed in all but two patients during their stay in the hospital. Antiviral therapy was administered in *n* = 17 (34%) of the patients, including mainly lopinavir/ritonavir (*n* = 12 (24%)) and, to a lesser extent, oseltamivir (*n* = 4 (8%)) and valganciclovir (*n* = 1 (2%)). An aminoquinoline derivative, chloroquine, was prescribed to *n* = 43 patients (86%). Antibiotic therapy was prescribed in all but four patients. Among antibiotics, macrolides and cephalosporins were most commonly used (*n* = 29 (58%) and *n* = 24 (48%), respectively). Other antibiotics used included glycopeptide antibiotics (*n* = 1 (2%)), carbapenem-type antibiotics (*n* = 10 (20%)), oxazolidinones (*n* = 3 (6%)), and tetracyclines (*n* = 1 (2%)). As many as *n* = 24 patients were treated with two different antibiotics and *n* = 2 patients with three. Anticoagulants were also given accordingly (*n* = 36 (72%)); among them, *n* = 4 were treated with oral anticoagulants.

The mean duration of hospital stay was 21.7 ± 14.5 days (median: 16.5; range: 4–57). There were *n* = 20 subjects (40%) who died during the hospitalization. Fifteen of them died within the first 15 days. No further fatal outcomes were reported during the 60 days period. On average, the non-survivors were significantly older than the recovered patients ((78.3 ± 9.7 years (median: 78; range: 66–99) vs. 72.4 ± 8.5 years (median: 70; range: 61–89), *p* < 0.05)). The duration of stay also varied between non-survivors and survivors ((15.4 ± 13.0 days (median: 11; range: 4–57) vs. 26.0 ± 14.1 days (median: 22.5; range: 6–56], *p* < 0.01)).

The comparison of clinical parameters and laboratory results between survivors and non-survivors are presented in Table 1 and Table 2. No differences in the frequency of typical symptoms were observed on admission (e.g., fever, dyspnea, and muscular pain). However, in non-survivors, there was a tendency to more frequent exacerbation of chronic somatic diseases (*p* = 0.06). Moreover, a similar pattern was observed for cognitive impairment (*p* = 0.09). There was also no difference for the COVID-19 clinical status (survivors: Mild, 9/30 (30%); Moderate, 15/30 (50%); Severe, 5/30 (17%); Critical, 1/30 (3%); non-survivors: 2/20 (10%), 9/20 (45%), 6/20 (30%), and 3/20 (15%), respectively; *p* = 0.15).

As far as comorbidities are concerned, notably, no significant difference was observed when comparing the frequency of cardiovascular diseases. However, when excluding hypertension and assessing heart diseases alone, they were observed to occur more commonly in non-survivors (*p* < 0.05). A nonsignificant tendency was also found for a higher frequency of diabetes in this group (*p* = 0.07). No other differences were observed.

Non-survivors had significantly higher WBC count, inflammatory markers (CRP, PCT, and IL-6), and LDH on admission, as compared to survivors (*p* < 0.05). RBC count, hematocrit, and hemoglobin were detected to be significantly lower in those who were discharged from the hospital (*p* < 0.01). There was no difference in the frequency of renal insufficiency (survivors: 9/30 (30%); non-survivors: 10/20 (50%); *p* = 0.23) and liver enzymes abnormalities (survivors: 5/21 (24%); non-survivors: 4/17 (24%); *p* = 0.72). Detailed laboratory results are presented in Table 2.

Furthermore, a significantly higher proportion of non-survivors required supplementary oxygen (16/20 (80%)) or mechanical ventilation (11/20 (55%)) during hospitalization, as compared to the survivors (12/30 (40%) and 3/30 (10%), respectively; *p* < 0.01). No other treatment differences were observed. Finally, before the hospitalization, a larger proportion of non-survivors were found to be functionally dependent as compared to the survivors (13/20 (65%) vs. 7/30 (23%); *p* < 0.05).

For the analysis of survival, only the four parameters of the medical history which differed according to the observed outcome were included in the multivariate model (age, functional status, concomitant heart diseases, and diabetes). In Cox univariate proportional hazard analyses, both in-hospital and 60-day survival were associated with heart disease morbidity and functional capacity. In multivariate models, functional capacity was the only factor affecting the outcome (Table 3). Kaplan–Meier curves of overall 60-day survival of functionally dependent and independent patients can be observed in Figure 1.

## 4. Discussion

As the COVID-19 pandemic continues to progress, it is vital that prognostic factors and risks associated with worsened course of infection continue to be identified and studied. Such factors are crucial to safeguard vulnerable populations, such as older adults. Varying clinical factors among a sample of 50 geriatric patients admitted to a dedicated COVID-19 hospital were analyzed in this retrospective study, with functional capacity, an unorthodox point of investigation, also examined.

In our study, typical symptoms of COVID-19 (increased body temperature, cough, dyspnea, and muscular pain) were relatively uncommon. As shown by other authors, such symptoms are more often present in younger subjects [6]. In older patients, instead of typical symptoms of infection, exacerbation of chronic conditions is frequently observed [17]. Correspondingly, in our study, one in five patients experienced exacerbation of chronic disease. Moreover, almost one in every four patients presented cognitive impairment, which is in line with the observation that such patients are more likely to develop COVID-19 [18]. On the other hand, they are also attributed a higher risk of delirium. Notably, delirium can be a presenting symptom even before fever or cough [19].

In regard to pharmacological treatment, aminoquinoline derivative—chloroquine—was administered as a standard part of treatment for a majority of our SARS-CoV-2 patients. Data regarding the risk–benefit ratio of aminoquinoline derivatives are conflicting [18,19], although it is worth mentioning that, in the period of March 2020–June 2020, the FDA (Food and Drug Administration) approved an Emergency Use Authorization for both chloroquine and hydroxychloroquine in adults who are hospitalized with COVID-19 and are unable to participate in clinical trials [20]. Alongside antivirals, antibiotic combinations were also utilized, mainly for prophylactic measures to prevent co-infection. Due to interstitial inflammation causing thrombosis in the course of COVID-19 infection [21], the use of anticoagulants has also been suggested as beneficial for the disease outcome [22]. As researchers continue to explore possible treatments and vaccines for the novel coronavirus, the abovementioned spectrum of drugs remained in the period of our study among the few options in treating COVID-19. Nonetheless, decisions on medications administered to elderly patients should be approached with caution, as adverse effects may outweigh benefits [23,24].

The mortality rate in the studied group, consisting of 50 patients, was observed to be substantial, at 40%. Nonetheless Sun et al. [11], who did a retrospective analysis of 244 older patients hospitalized due to COVID-19, observed a mortality rate of almost 50%. Moreover, one hospital, which was included in a multicenter European observational cohort study, revealed a mortality rate above 40%, despite the inclusion of relatively younger patients ≥ 18 years old (while adults at the age of ≥60 years have been shown to experience more severe mortality rates than other age groups [4,6,20,21]). A relatively high mortality rate in our study seems to reflect the health status of the hospitalized elderly patients. Most of them, importantly, were hospitalized due to comorbidities and transferred to a monomial hospital after their COVID-19 diagnosis (among them, there were also palliative care patients). Hence, these transferred patients could be perceived as particularly vulnerable and their survival rate was lower compared to those who were hospitalized directly in the dedicated COVID-19 hospital.

We found that non-survivors more commonly exhibited symptoms which are typical for old age, such as exacerbations of chronic diseases or cognitive impairment. Recent publications suggest the possibility of decreased COVID-19 survival as a result of exacerbations of pre-existing diseases, partly due to the effect of pneumonia developed during infection [22,23]. Moreover, cognitive impairment in the course of dementia was shown to be related to higher mortality [19]. Further studies on larger patient groups are necessary to confirm our observations.

As far as comorbidities are concerned, non-survivors had heart diseases more often than survivors. Wu and colleagues, when analyzing over 44,000 cases of COVID-19 in Wuhan, China, noted roughly 4.5 times higher case fatality rate in patients with heart diseases compared to the general population (10.5% and 2.3%, respectively) [24]. Guzik et al. further described heart-related conditions as detrimental risk factors for patients infected with the novel coronavirus [25]. In accordance with existing literature [26,27,28], it should be noted that comorbid hypertension was not a point of contention when distinguishing between survivors and non-survivors in our study, as it was found to be of insignificance when comparing the two groups (*p* = 1).

The results of other studies prove the association between diabetes and increased mortality [29]. We found a tendency of decreased survival rate; however, a more comprehensive analysis is needed to assess the significance of this factor. Non-survivors in our study had higher levels of inflammatory markers (WBC, procalcitonin, CRP, and IL-6) as well as lower anemia related parameters (RBC, hemoglobin, and hematocrit), which concurs with current publications [30].

The most important parameter, which increased the survival rate of our patients, was independence in daily activities. It was more important than age, diabetes, and even heart diseases. We decided to include this parameter since earlier research demonstrated a worse course of infectious diseases in older persons with limited independence. We were, however, due to sample size, only able to show that independent subjects had a three-fold higher chance of 60-day survival than their dependent counterparts. Still, our study reinforces a discussion on the need for incorporating assessment tools for functional status and frailty when developing strategies for risk evaluation in older individuals with COVID-19. To the best of our knowledge, we are among the first to pay attention to this factor. One study which incorporated a similar parameter, frailty, stated its use as a predictor of outcome in COVID-19 patients as better than comorbidities or age alone [31]. We observed no difference in the clinical status of survivors and non-survivors, which may be related to the fact that the assessment took place on admission when the individual patients presented various stages of the disease.

A decline in functional capacity has been shown to have a direct correlation with a lack of physical exercise and increased periods of stagnation in elderly patients [32]. Inactive lifestyles are said to potentially increase the risk of heart disease, diabetes mellitus and obesity by 100% [33], which are all detrimental to the prognosis of COVID-19 [34]. Furthermore, it has been shown that an absence of regular physical exercise may increase the frequency of infections in the respiratory tract and the fatality rate of patients with pneumonia and influenza [35]. A similar logic also applies to a host of other common diseases, both acute and chronic [36,37,38]. Applying this rationale to our findings, it may be suggested that the worsened clinical outcome for geriatric COVID-19 patients with poor functional capacity may be a result of the dependence on carers to perform everyday tasks, and an incapacity to perform regular physical exercise. This is further exacerbated by the tendency for individuals with poor functional capacity to experience an inability to feed self, leading to a poor diet [39]. In essence, deterioration of functional capacity may result in a more sedentary lifestyle, with unhealthy or insufficient diet that in turn leads to worsening of immunity and development of comorbidities that lead to adverse outcomes in COVID-19.

Some notable limitations were experienced in the presented study. Due to the novelty of the disease, sample size and follow-up duration of patients were restricted to 50 patients and 60 days, respectively. The results should be interpreted with caution as limited sample size presents limitations on analysis adjustments and less room for confounder adjustments. Confounding variables found insignificantly different in initial analysis were not adjusted for in further analysis (this includes pulmonary and renal disease that may be influential in larger samples and should be taken into consideration). The sample was also largely homogenous (white Caucasian origins) and may therefore not be fully reflective of disease presentation in other populations. Nonetheless, the presented results may act as preliminary to more meticulous studies. Larger samples of patients with more prolonged monitoring of outcome can assist in providing a more comprehensive insight into the relationship between certain prognostic factors and COVID-19 outcome in older individuals. Additionally, standardizing certain assessments, such as applying well established functional capacity scores, may shed light into relevant associations to a greater detail.

On the other hand, our study has important strengths; we present the Central-Eastern European perspective of COVID-19. Our study, as one of few, presents not only the in-hospital survival of patients but also 60-day survival. We were thus able to demonstrate that 75% of deaths took place no later than on the 15th day of hospitalization. Henceforth, surviving the first 15 days increases chances for a positive outcome considerably. Additionally, our paper stresses the necessity to incorporate the geriatric perspective into the evaluation of older subjects hospitalized for COVID-19, as the assessment of functional capacity showed to be the most important prognostic factor.

## 5. Conclusions

Functionally dependent older adults infected with the SARS-CoV-2 virus were more likely to face mortality in the 60-day period following diagnosis, as compared to independent older patients. Exploring geriatric approaches, such as assessment of functional capacity, may assist in constructing comprehensive prognoses in the elderly population with COVID-19.

## Figures and Tables

**Figure 1 ijerph-17-08362-f001:**
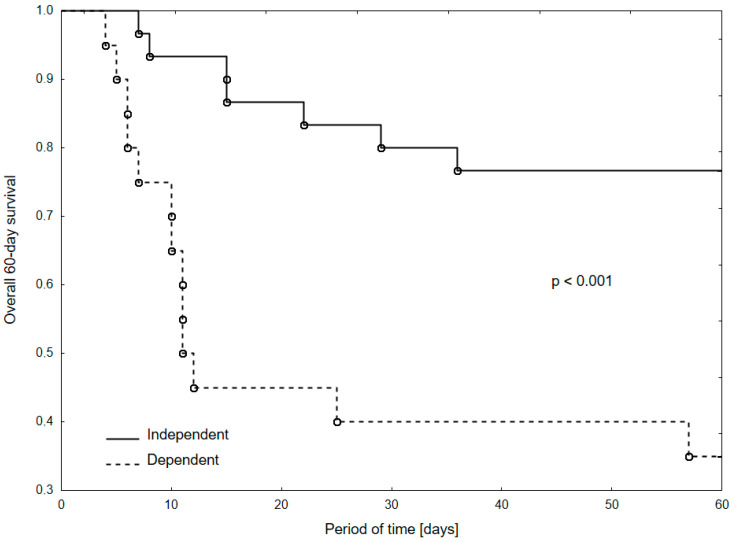
Kaplan–Meier curves of overall 60-day survival model in functionally independent vs. dependent patients.

**Table 1 ijerph-17-08362-t001:** Symptoms and comorbidities of the first 50 elderly COVID-19 patients in Poznan, Poland (survivors vs. non-survivors).

	Total (*n* = 50)	Survivors (*n* = 30)	Non-Survivors (*n* = 20)	*p*-Value
Age	74.8 ± 9.4 (72.5; 61–99)	72.4 ± 8.5 (70; 61–89)	78.3 ± 9.7 (78; 66–99)	<0.05
Gender				
Male	35 (70.0%)	23 (76.7%)	12 (60.0%)	0.23
Female	15 (30.0%)	7 (23.3%)	8 (40.0%)	
Symptoms				
Fever *	29 (58.0%)	20 (66.7%)	9 (45.0%)	0.15
Cough	14 (28.0%)	9 (30.0%)	5 (25.0%)	0.76
Dyspnea	17 (34.0%)	10 (33.3%)	7 (35.0%)	1.0
Muscular pain	3 (6.0%)	1 (3.3%)	2 (10.0%)	0.56
Exacerbation of chronic diseases	10 (20%)	3 (10%)	7 (35%)	0.06
Cognitive impairment	11 (22%)	4 (13.3%)	7 (35%)	0.09
Comorbidities				
Cardiovascular diseases	40 (80.0%)	23 (76.6%)	17 (85.0%)	0.72
Hypertension	30 (60.0%)	18 (60.0%)	12 (60.0%)	1.0
Heart diseases	26 (52.0%)	11 (36.7%)	15 (75.0%)	<0.05
Diabetes	19 (38.0%)	8 (26.7%)	11 (55.0%)	0.07
Chronic Obstructive Pulmonary Disease	7 (14.0%)	5 (16.7%)	2 (10.0%)	0.69
Renal Dysfunction	8 (16.0%)	3 (10.0%)	5 (25.0%)	0.24
Liver Dysfunction	3 (6.0%)	1 (3.3%)	2 (10%)	0.56
Malignancy	6 (12.0%)	4 (13.3%)	2 (10.0%)	1.0
Dementia	2 (4.0%)	0 (0.0%)	2 (10.0%)	0.16
Stroke	11(22.0%)	6 (20.0%)	5 (25.0%)	0.74

* Fever was diagnosed as axillary temperature > 38.0 °C.

**Table 2 ijerph-17-08362-t002:** Laboratory results (COVID-19 survivors vs. non-survivors).

Lab Tests	Normal Range	Total (*n* = 50)	Survivors (*n* = 30)	Non-Survivors (*n* = 20)	*p*-Value
White Blood Cells (WBC) (×10^3^/µL)	4.0–11.0	8.4 ± 4.5 (7.3; 2.3–29) *n* = 50	7.1 ± 2.7 (7; 2.7–12.5) *n* = 30	10.2 ± 6.0 (8.8; 2.3–29) *n* = 20	<0.05
Red Blood Cells (RBC) (×10^6^/µL)	4.50–6.10	4.2 ± 0.7 (4.2; 1.9–5.6) *n* = 50	4.4 ± 0.6 (4.4; 3.2–5.6) *n* = 30	3.9 ± 0.7 (4; 1.9–4.8) *n* = 20	<0.01
Hemoglobin (g/dL)	14.0–18.0	12.3 ± 2.1 (12.4; 6.6–17.3) *n* = 50	13.1 ± 1.9 (13.1; 9.1–17.3) *n* = 30	11.1 ± 1.8 (11.2; 6.6–14.1) *n* = 20	<0.001
Hematocrit (%)	38.0–55.0	35.9 ± 5.9 (35.1; 18.5–54.3) *n* = 49	38.0 ± 5.4 (36.8; 28.5–54.3) *n* = 29	32.9 ± 5.4 (33.4; 18.5–40.2) *n* = 20	<0.01
Platelets (×10^3^/µL)	30–440	240.8 ± 108.3 (223.5; 55–502) *n* = 50	238.4 ± 101.8 (223.5; 98–502) *n* = 30	244.4 ± 120.1 (225; 55–477) *n* = 20	0.91
Lymphocytes (×10^3^/µL)	1.0–4.0	1.3 ± 0.6 (1.2; 0–2.9) *n* = 33	1.3 ± 0.7 (1.2; 0–2.9) *n* = 23	1.3 ± 0.4 (1.2; 0.8–2.2) *n* = 10	0.74
Neutrophils (×10^3^/µL)	1.5–7.7	5.1 ± 3.2 (4.6; 0.1–14.3) *n* = 33	4.4 ± 2.1 (4; 1.4–9.5) *n* = 23	6.7 ± 4.6 (6.8; 0.1–14.3) *n* = 10	0.12
Urea (mmol/L)	3.6–7.1	9.0 ± 5.1 (8; 3.1–26.2) *n* = 33	7.6 ± 3.8 (6; 3.1–15.4) *n* = 18	10.7 ± 6.1 (9.7; 3.2–26.2) *n* = 15	0.16
Lactate Dehydrogenase (U/L)	125–220	368.8 ± 145.6 (352.5; 141–660) *n* = 36	309.4 ± 110.1 (284.5; 141–560) *n* = 22	462.1 ± 14.9 (430.5; 193–660) *n* = 14	<0.01
CRP (mg/L)	<5.0	93.2 ± 86.1 (66; 1–384) *n* = 50	75.3 ± 75.3 (45.8; 1–290.5) *n* = 30	120.0 ± 96.0 (78.5; 33.2–384) *n* = 20	<0.05
PCT (ng/mL)	<0.10	1.0 ± 4.8 (0.1; 0–29.8) *n* = 38	0.1 ± 0.1 (0.1; 0–0.4) *n* = 19	1.8 ± 6.8 (0.3; 0–29.8) *n* = 19	<0.01
IL-6 (pg/mL)	1.5–7.0	128.2 ± 273.1 (54.7; 1.5–1592) *n* = 36	63.6 ± 113.1 (29.7; 1.5–500) *n* = 20	208.9 ± 381.2 (86.9; 6–1592) *n* = 16	<0.01

**Table 3 ijerph-17-08362-t003:** Cox proportional hazards regression of factors affecting in-hospital and overall 60-day mortality (60 days from initial hospitalization, including after discharge period) among the first 50 elderly patients (≥60 years) hospitalized due to COVID-19 in Poznan, Poland.

Assessed Variables	In-Hospital Survival		60-Day Survival	
	Univariate Model	Multivariate Model	Univariate Model	Multivariate Model
Age [≥75 yrs vs. 60–74 yrs]	1.73 (0.69–4.31); *p* = 0.24	-	2.11 (0.86–5.18); *p* = 0.10	-
Functional Capacity [Dependent vs. Independent]	2.97 (1.16–7.59); *p* = 0.02	2.36 (0.90–6.23); *p* = 0.08	4.18 (1.66–10.54); *p* = 0.002	3.34 (1.29–8.63); *p* = 0.01
Diabetes [Yes vs. No]	1.63 (0.66–4.03); *p* = 0.29	-	2.35 (0.97–5.67); *p* = 0.06	-
Heart Disease [Yes vs. No]	3,05 (1.10–8.47); *p* = 0.03	2.42 (0.84–6.94); *p* =0.10	3.49 (1.27–9.63); *p* = 0.02	2.61 (0.92–7.39); *p* = 0.07

Data are shown as hazard ratios (HR) with 95% confidence intervals. “-“ refers to restrictions to perform Cox proportional regression analysis due to limited sample size.

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
