# Peer review of "Clinical Characteristics and Survival Analysis in a Small Sample of Older COVID-19 Patients with Defined 60-Day Outcome"

_ijerph, 2020, doi:10.3390/ijerph17228362_

Round 1
Reviewer 1 Report
Even if the study cohort is small and the study retrospective, I was surprised by the quality of this paper on an emerging topic.
The paper brings a new and interesting geriatric light in the field of the Covid pandemics.
Personally, I propose you to accept this paper without modification
Author Response
We would like to thank the reviewer for their kind words - no adjustments requested
Reviewer 2 Report
The authors of this article catched an important research question. However, regarding the small sample size the conclusions of the manuscript may be questionable. Especially as crude hazard ratios are presented. The manuscript could be improved by some more explicit explainations especially in the statistical analyses and less detail in some explainations concerning the results section. The finishing lines of the discussion are very well thought of and formulated.
Title: The abstract suggests more in depth analyses to the topic than the title does (Survival analyses). The authors may think about to revise the title.
Abstract:
Overall: The essence of the paper is not easily to understand while just reading the abstract. Some important information is missing and the authors should consider to condense their results to the essence, in order to be really clear about the topic and the impact of this research.
Line 24 „remains amongst“ necessary. Suggestion: is one of the most vulnarable…
Line 25 The rational of this study could be more clearly stated. This study is certainly important since not much is known about this group.
Line 27-29: Rewording might be necessary to catch the point of this sentence.
Line 29: It is advisable to add some information what kind of clinical data was assessed.
Line 32: How many survivors/non survivors were included? Each group =25?
Line 38: „functional dependence“ requires some more definition or introduction.
Line 38; what is the meaning of bivariable cox regression? Please explain what was done exactly.
Line 39: Please give confidence intervals.
Line 39/40; be more specific. What is the meaning of „poor outcome“? A solid Interpretation of the HR estimate is missing.
Line 40-42: It is unclear for the reader which prognosis is meant.
Keywords: For the age group under study the authors may wish to chose the term „Elderly“ instead of „Older Adults“.
Introduction:
The introduction is well written and gives the reader a good overview of the current knowledge and the necessity for the topic. However, the specific characteristics that will be studied (e.g. the mentioned „Clinical data“ in the abstract) remain to be introduced somehow. Also the rational for chosing 60 days of survival remains unclear.
Line 50: Please erase the word astonishing.
Line 75: „concoct“ replaced by „conduct“ or „develop“?
Methods:
The methods section suffers from an unprecise description of those methods that were applied. The parapgraph could be devided into sub-titles. A statement about written informed consent by the participants is missing. Regarding the laboratory analyses the name of the analyzing lab and quality measures are necessary to be included.
Line 85: It is unclear which first 50 patients were included. A date should be added.
Line 86: The term retrospectively remains unclear a short sentence what exactly was analysed would be of help.
Line 87: The brand of the test should be added.
Line 89-99: Was there any standardisation for testing?
Line 100: Erase the term analyzed. The analyses is population based not individual based.
Line 104-108: Were the measurements taken according to a standardized porcedure?
Line 108: How was cognitive impairment measured?
Line 109- 112: Who did these assessments, what was the basis? Medical records?
Line 113: Who assigned the COVID-19 status?
Line 118 ff.. Who did the laboratory analyses and which methods were used?
Line 122 and 123: a reference is missing for the chosen cut-offs.
Line 124-126: who did that?
Line 146: „The scoring“ suggests a grouping within a certain range. However, this remains unexplained. This should either be explained or a reference should be given.
Statistical analyses:
The used statistics are common procedures to test for significane. The applied Cox regression analyses could be elaborated in more detail.
Line 162: „In-hospital and 60-day survivals“ probably the authors compared the hospital and at home 60 day survival. That needs to be described more clearly.
Line 163-164: Cox proportional hazards regression requires some more information. Did the authors consider any confounder adjustment? Some explaination of bivariate Cox regression or a reference is adbvisable. A definition of retrospective design is missing.
Line 165-166: Sentence two is redundand.
Results:
The results section is in some points too elaborated and in other points not thoroughly written. Important information regarding the tables that do support the main text is missing.
Line 168: (n=29 - 58%) should be (n=29 (58%)).
Line 176: While thirty patients. Please change in n=30 patients.
Line 177: Please add n=
Line 191-200: The Explanation regarding medical subscriptions are very detailed. A Table may be advisable. The text should incorporate the most imoportant factors related to the analyses.
Line 209: „Are presented in“ instead of „is presented“.
Line 231: More explicit title of the table is required. Footnotes are missing or a definition of e.g. fever.
TABLE line 241-43: The title does not clearly state what the analyses is about. E.g. what is presented, which study is it. How big is the sample size. The table should be prepared in order to be understandable also without having the manuscript at hand. It is unclear if and which reference group was chosen, that should be incorporated in the table as Ref. HR 1.0. What is the sample size of each analyses? What does“ – „means in the table? I suppose calculations were not possible due to the sample size? A legend for this Table is required. It remains unclear what the authors did in bivariate analyses.
FIGURE line 242: An explanation which model is presented and which adjustements were performed is missing.
Discussion
Generally the discussion could be written more to the point. The last paragraph of the discussion is very well written and states the importance of the current analyses.
Line 251: The retrospective design remains to be clearified. Which characteristics were available prior to death? Age, disease status etc. was assessed during submission to the hospital. Death occurred after admission to the hospital. However, this would qualify the analyses to be prospective. Clarification is recommended.
260-272: This paragraph suggests which medication could support COVID-19 treatment. However, it was not mentioned that this was the purpose of the manuscript.
273-78: The total number of participants is necessary to enable interpreation of these results.
Line 289: The small sample size is mentioned here for the first time. This aspect needs further consideration. A Power calculation is recommended.
Line 295-96: The authors should elaborate the meaning of the sentence „it should be noted that hypertension was not a point of contention when distinguishing between survivors and non-survivors“
Line 302ff.: This paragraph misses the discussion wheather the factor „functional status“ or rather related factors to functional status that may lead to physical dependency is relavant for the association between COVID-19 and survival.
Line 315: Discussion of confounder adjustment is missing. Also the limitation of unstandardized assessments (see methods section) is required. The authors should discuss the expected impact of these limitations on the analyses.
Author Response
We would like to thank the reviewer for their concise and constructive feedback on the manuscript. Detailed adjustments and response to all comments raised by the reviewer can be found in the attached document. We are confident the changes implemented are of value to the paper.

Reviewer 3 Report
The older population remains amongst the most vulnerable to experience adverse outcomes of COVID-19 pandemic. So the aim of the study was to characterize the difference between survivors and non-survivors among hospitalized older adults, to further investigate how those with increased risk of poor outcome can be identified as early as possible. The paper presents the Central Eastern European perspective of elderly with COVID-19 for the first time.
The paper has a clear friendly structure (Introduction, Method, Results, Discussion and Conclusion). The topic is crucial and very time-sensitive, and the article addresses important issues regarding the most predictive factors of poor outcome in older people with COVIC-19. The material is original and the research methods are relevant and sufficiently described. The manuscript is distinguished by a good in-depth analysis of the results and an extensive discussion including a limitation section. The text is supplemented by 3 tables, one figure and 37 current references.
Minor issues:
- In the discussion, in the limitation section, the description of the epidemiological situation due to the COVID-19 pandemic is no longer valid: ‘For example, we had neither a phase of rapid increase of infections nor high mortality in general. But, in Poland, we still observe a steady, slight increase in morbidity and stable mortality in the range between 10 and 30 deaths per day’. In Poland, we are currently observing a rapid increase in morbidity and mortality. The worldwide pandemic situation is changing so rapidly that some statements may be out of date at the time of publication. Therefore, I propose to delete such statements and be very careful in assessing the epidemiological situation.
Author Response
We would like to thank the reviewer for their positive feedback. As advised, the minor adjustment requested has been implemented as follows:
In the discussion, in the limitation section, the description of the epidemiological situation due to the COVID-19 pandemic is no longer valid: ‘For example, we had neither a phase of rapid increase of infections nor high mortality in general. But, in Poland, we still observe a steady, slight increase in morbidity and stable mortality in the range between 10 and 30 deaths per day’. In Poland, we are currently observing a rapid increase in morbidity and mortality. The worldwide pandemic situation is changing so rapidly that some statements may be out of date at the time of publication. Therefore, I propose to delete such statements and be very careful in assessing the epidemiological situation.
Lines 320-324 were removed as advised.
Round 2
Reviewer 2 Report
The manuscript generally improved. However, the authors may wish to include the survival analyses as being an “add on” to this manuscript instead of being a main outcome. During the first run I suggested to change the title of the paper. However, considering the sample size and the limited possibility of adjusting the results for potential confounders, I would like to advise the authors to present the HR with more caution. The survival analyses may stay part of the manuscript, but requires still much more cautious interpretation. This has partly been improved by the authors in line 358” Nonetheless, the
presented results may act as preliminary to more meticulous studies.” Having performed survival analyses as an additional analyses should be included in the abstract if the authors wish to keep those results included.
Title: may require final changes after having revised the manuscript.
Line 38: Multivariate cox regression is misleading. Was there any other variable included in the model despite Exposure and outcome? If not, this line should include that HR were unadjusted.
Line 42: please change this sentence into: Elderly with COVID-19 who required assistance in core… had a 3 times increased risk to experience mortality, as compared to….
Line 110: The first Covid-19 patient (aged xxx), instead of older patient.
The first analysed older patient was hospitalised in the Jozef Strus Hospital on the 12th of March, 2020 and the last – on the 5th of April, 2020.
Line 121: Please clearify what patient examination included or add a reference, explaining the procedure in detail.
‘Cognitive impairment was determined based on patient examination by attending clinicians upon admission to the hospital.’
Line 126: In combination or was there a priority list?
‘Infectious disease specialists or internal medicine physicians, and nurses' reports included in hospital medical records, were the basis for assessments.’
Line 140: please add in the central labs of the monomial hospitals.
Line 176: I suppose ® should be included behind the used statistical package.
Line 185: It is still unclear if adjustments were performed and which covariates the authors considered.
Line 191: Information on demographics is missing. Information on person years accumulated by the patients is missing.
Line 191 to 227 may be considered to be condensed.
Line 253: a legend (number or sign) is missing.
Line 285: does this mean confounder adjustment took not place? Please clarify.
“All laboratory analysis were based on blood samples retrieved from patients and tested in the central hospital lab“
Line 265: add Cox proportional regression analyses to the following sentence:
” […] refers to restrictions to perform regression analysis due to limited sample size”
Line 268: Please add physical independent vs. dependent
“Kaplan-Meier curves of overall 60-day survival model in independent vs. dependent patients.”
Line 356: The authors should state which confounding variables are of concern that could not be taken into account for the current analyses. Here they could also argue if the sample was homogeneous in some aspects which would limit the thread on the results.
Table 1: This table or a baseline table should also include age and sex as well as the other known demographic characteristics. The association is likely biased by age and sex. However, if the sample is homogeneous confounding by age and sex is less likely. This information would pretty much help the interpretation of the results and is also an important information for the discussion section.
Functional capacity should be added here. Main analyses should be the descriptive analyses with the add-on analyses: Cox regression for those outcomes which turned out to be of relevance.
Table 2: the authors may wish to add COVID-19 survivors vs. non-survivors.
Table 3: information regarding the multivariate nature of this model is missing here and in the methods section.
Please add: […] among the first 50 elderly patients …
Is there a purpose for age being written in a different style?
General remarks: The authors may wish to consider a sensitivity analyses stratifying the data by e.g. age, very young and very old and performing significance testing thereafter. This would help to limit confounding by e.g. age.
Author Response
Once again, we would like to kindly thank the reviewer for their constructive feedback. Adjustments have been made accordingly and are detailed in the attached file.
